# Current Insights into Monitoring, Bioaccumulation, and Potential Health Effects of Microplastics Present in the Food Chain

**DOI:** 10.3390/foods9010072

**Published:** 2020-01-09

**Authors:** Leonard W. D. van Raamsdonk, Meike van der Zande, Albert A. Koelmans, Ron L. A. P. Hoogenboom, Ruud J. B. Peters, Maria J. Groot, Ad A. C. M. Peijnenburg, Yannick J. A. Weesepoel

**Affiliations:** 1Wageningen Food Safety Research, Part of Wageningen University & Research, P.O. Box 230, 6700 AE Wageningen, The Netherlands; meike.vanderzande@wur.nl (M.v.d.Z.); ron.hoogenboom@wur.nl (R.L.A.P.H.); ruudj.peters@wur.nl (R.J.B.P.); maria.groot@wur.nl (M.J.G.); ad.peijnenburg@wur.nl (A.A.C.M.P.); yannick.weesepoel@wur.nl (Y.J.A.W.); 2Aquatic Ecology and Water Quality Management Group, Wageningen University & Research, P.O. Box 47, 6700 AA Wageningen, The Netherlands; bart.koelmans@wur.nl

**Keywords:** microplastic, occurrence, human health, analytical methods, quality control

## Abstract

Microplastics (MPs) are considered an emerging issue as environmental pollutants and a potential health threat. This review will focus on recently published data on concentrations in food, possible effects, and monitoring methods. Some data are available on concentrations in seafood (fish, bivalves, and shrimps), water, sugar, salt, and honey, but are lacking for other foods. Bottled water is a considerable source with numbers varying between 2600 and 6300 MPs per liter. Particle size distributions have revealed an abundance of particles smaller than 25 µm, which are considered to have the highest probability to pass the intestinal border and to enter the systemic circulation of mammals. Some studies with mice and zebrafish with short- or medium-term exposure (up to 42 days) have revealed diverse results with respect to both the type and extent of effects. Most notable modifications have been observed in gut microbiota, lipid metabolism, and oxidative stress. The principal elements of MP monitoring in food are sample preparation, detection, and identification. Identified data gaps include a lack of occurrence data in plant- and animal-derived food, a need for more data on possible effects of different types of microplastics, a lack of in silico models, a lack of harmonized monitoring methods, and a further development of quality assurance.

## 1. Introduction

Current food production is characterized by a lot of food packed in plastic materials, both in by-products and in waste flows. Initiatives are being taken to reduce the release to the environment. In 2015, the European Commission (EC) presented a comprehensive plan for a circular economy [1]. In 2018, the EC launched a strategy for reducing plastic in the context of a circular economy [2,3]. A range of national initiatives for the reduction of the use of plastic have been installed or are in preparation [4]. Methods for the reduction of MPs are pending (i.e., filtration from wastewater effluent [5]); however, insufficient data have been reported to indicate trade-offs and effectiveness of these methods for food matrices in a meaningful way. The extensive use of plastics has resulted in the presence of microplastics (MPs) in the food chain and exposure to consumers. Studies have reported the presence of MPs in water, seafood (including shellfish), sugar, honey, and beer [4]. Research into human feces has shown the presence of an average of 20 MP particles in 10 g of material from eight people from eight countries [6]. It is therefore important to collect and review information on both exposure to and potential effects in humans. A distinction has been made between primary and secondary MPs [7]. The first category refers to particles produced for a specific function, e.g., skin treatment, molding, or cosmetics [8], and the second to particles resulting from the wearing and abrasion of e.g., packaging material, clothes, bottles, and tires [9,10,11]. For the management of sources, the distinction of these categories is very important, but in the current review the origin is considered less relevant, since regardless of the source, all MPs may enter the food chain [10].

Microplastics are usually defined as particles with a size smaller than 5 mm for their largest dimension [7,9,12,13]. In general, particles 1–2 mm in size and larger can be visually detected and extracted by eye and (therefore) quantified based on mass. This has become a daily practice in the monitoring of former foodstuffs for use in animal feeds [14]. Particles smaller than 1 mm can only be detected microscopically and therefore quantification is limited to counting. In this review, particular attention is paid to the presence and effects of MP particles smaller than 1 mm (or 1000 µm, known as small microplastics) [8,15], because this is an area where relatively little is known regarding the field of food safety. Furthermore, particles exceeding this size are expected to be incapable of passing the gut wall. This focus is in line with the recommendations of the European Food Safety Authority and of Hartmann et al. [9,10]. The lower limit is chosen here as 1 µm, although other lower limits for size range have been chosen in the literature. In several reports a lower limit of 100 nanometers (nm) has been adopted or mentioned, primarily because this has been chosen to be the upper limit of nanoplastic [9,13,16]. However, detection and identification of particles below the size of a few µm require other strategies than those used for larger particles. As an example, identification using FTIR is only achievable for particles over 10 µm [17,18]. Due to the wavelength of visible light, the lower limit for visual detection using standard microscopes is theoretically 0.2 µm, but is over 1 µm in practice. Two review papers on monitoring methods have also shown that the lower limit of the particle size distribution (PSD) has been 1 µm or larger in the reviewed studies [19,20]. Similarly to the current review, several other studies have also chosen to use a lower limit of 1 µm in their definition [12,15,21,22,23,24], or a limit even larger than 1 µm [10]. The lower limit of the PSD as applied in a survey is an essential element in study design. In the view of a continuous degradation of large particles into smaller particles, the smallest size class can be assumed to be the most abundant. The smaller the minimum size of the PSD, the more particles can be assumed to be found. On the other hand, aggregation of nanoparticles is mentioned in the definition as used by the European Food Safety Authority (EFSA) [9]. This might finally result in conglomerates in the micro-size range, as has been observed for ambient black carbon conglomerates at the fetal side of the placenta [25]. The distribution kinetics of microparticles in terms of Brownian motion versus Buoyance force is shifted at sizes of 1–1.5 µm for polyethylene (PE) and polystyrene (PS) [26], which means that besides different analytical approaches the evaluation of bioaccumulation and effects might also be different.

In 2016, EFSA published a statement on the status quo with respect to MPs [9]. It was recommended to pay attention to the development of validated monitoring methods and to the collection of data on the occurrence of MPs in feed and food products, and to gain knowledge on the possible toxicity of MP particles. These recommendations have been generally repeated in more recent overviews [12,13]. The present study has intended to review the presence and concentrations of MPs in the food production chain and to evaluate their possible effects by discussing results published after the reviews of EFSA and the Food and Agricultural Organization of the United Nations (FAO) [7,9]. As far as possible, information on occurrence and concentrations in soil, water, and food commodities will be given in counts in concordance with the remarks of EFSA and FAO on analytical aspects. These results will be evaluated in the present state of quality assurance and control of detection and identification methods. In addition to the reviewed documentation, relevant knowledge gaps will be identified.

## 2. Occurrence and Concentrations in the Food Production Chain

The food production chain is interpreted in this review in a broad sense. Production of food starts with soil quality and the use of fertilizers. Plants respond to environmental parameters, including microparticles like MPs that might be present in soil. Exchange of microparticles and contaminants between sediment and water influences water quality and hence the living environment of aquatic animals. The presence of MPs in drinking water adds to the possible occurrence of MPs in food products as part of daily diet. The several available routes for MPs for entering the food chain have been summarized by Karbalaei et al. [4]. The different elements of food production and consumption will be addressed in the following section.

The first depot of MPs which can enter the food chain can be found in sediments. Ranges in abundance (particles kg^−1^ sediment dry weight (dw)) have been reported for, e.g., beaches in South Korea (0.9–4463; 50–5500 µm fiber length), Mexico (0–4800; 4.3–4500 µm), and the coast of Spain (101–897; 63–2000 µm) [12]. In sediment along the Rhine between Mainz and Bingen (middle Rhine area) and some tributaries, amounts of MPs have been found up to 1 g kg^−1^ (0.1%) or 4000 particles kg^−1^ in the size range 63 to 5000 µm [27]. The most common plastic types observed have been PE, polypropylene (PP), and PS, which together were found to comprise 85% (*w*/*w*) or 92% (number of particles) of all MPs as identified in the sediments in this case. PS comprised 54% (number of particles), and PP was most abundant in terms of weight (49% *w*/*w*) [27]. Numbers between 1400 and 4900 particles kg^−1^ dry sediment weight with a PSD of 10–5000 µm have been reported for three sampling locations along the rivers Rhine and Meuse [28]. Quantities of 100 to 3600 particles kg^−1^ dry weight have been found in sediments along the Dutch North Sea coast [28]. Sediments along the Brisbane river (Australia) have revealed the presence of 10 to 520 particles kg^−1^, recalculated to 0.18 to 129.20 mg kg^−1^ sediment. Primarily PE, polyamide (PA), and PP were found in this instance. There was a positive relationship between abundance and wet versus dry season, and with dispersal hotspots, both in terms of counts and in weight [29]. This study does not state a PSD. There is a good relationship between numbers and weight across the seasons, indicating a comparable PSD. However, He et al. have discussed the difficulty of comparison among different studies for the different approaches applied [29].

Samples of fertilizer from 13 different plants for digesting or composting bio-waste were reported to contain counts of MPs between 0 and 146 particles kg^−1^, with sizes found between 1 and 5 mm. Samples from three out of these 13 plants showed numbers higher than 11 particles kg^−1^. A fourteenth plant contained 895 particles kg^−1^. The MP particles consisted of spheres, amorphous structures, and fibers. The most abundant types were styrene-based polymers (46%), polyethylene terephthalate (polyester, PET, 19%), PE (14%), and PP (8%). Fragments smaller than 1 mm were found but were not included in the final results [30]. The consequences of MPs in soil on plant growth have been largely underexplored. Effects on plant growth, either positively or negatively, can be expected based on current knowledge in soil science and on the physicochemical properties of both native soil particles as well as anthropogenic particles [31]. Especially the presence of fibers in soil will result in modified structure and bulk density of the soil and in better aeration. Direct uptake of MPs by plants via their root system has not been assumed. In contrast, nanoplastics (NPs) have been considered to be capable of being transferred from soil to roots, which might be an issue in view of the degradation of MPs into NPs [31]. After exposure of a brown seaweed (*Fucus vesiculosus*) to PS particles (size 20 µm) at a concentration of 2.65 mg liter^−1^ (597 particles mL^−1^), a number of 3.99 on average (range 0.9–6.1) PS particles mm^−2^ seaweed surface were found. After washing a reduction of 94.5% was achieved [32]. These results indicate a certain ability of MPs to strongly bind to the surface of seaweed, but systemic intake by seaweed remains undocumented.

There is very little information on the occurrence of MPs in food. The majority of data applies to food from marine environments [4,9,12,16]. Shellfish and small fish might give the most explicit exposure via food, since the gastro-intestinal (GI) tract of some species is consumed as part of whole animals and generally the concentrations of MPs are highest in the GI tract. In contrast, vertebrate animals, including most fish species, are prepared by removing the GI tract before consumption, in addition to other processes of preparation such as the removal of gills, skin, scales, and fins. Several reviewed studies have reported the presence of MPs in fish, shrimps, and bivalves at relatively low levels in the higher part of the MP size range [9].

A recent review on the human intake of MPs from food gave an estimate of 39,000 to 52,000 particles per year depending on age and gender [33]. This estimate was based on an evaluation of 26 studies with different sorts of food products, which together accounted for 15% of the food products consumed in the United States, covering seafood, water, sugar, salt, and honey. For cereals, vegetables, and meat no reports were found for this review. A general PSD distribution was not provided because this was dependent on the design of each of the studies [33]. One indication has been given from an included study on bottled water [34]. This study used a lower limit of the PSD of 6.5 µm, which has been indicated by Cox et al. to be smaller than most other studies [33]. Tap water has been assumed to account for around 10% of this exposure (4000 particles per year). Consumption of exclusively (mineral) water from bottles would result in an additional exposure of approximately 90,000 MP particles per year. It has been indicated that the estimates should be seen as an underestimation [33]. A wider study has been carried out by the European Union’s (EU) Joint Research Centre comprising an overview of approximately 200 papers selected from an initial set of 4000, covering 201 edible animal species; marine fish species (164) were primarily covered, with some additional food products such as sea salt, sugar, honey, beer, and water also covered [13]. Only results for one terrestrial animal species (chicken) were reported. These chickens, collected from a Mexican village where the animals had to search for their feed in a highly polluted area, appeared to contain MPs. Neither a precise nor global level of exposure came out of this overview study. Instead, it was concluded that exposure to humans is present along a variety of routes which has also been pointed out by Karbalaei et al. [4]. Some reviews present tables with actual data on the presence of MPs in marine and freshwater sediments and aquatic animals, and on effects [4,12]. Large differences have been found in MP concentrations in sediment and aquatic animals. An overview of MP concentrations in fish has revealed in general few MP particles per specimen, namely, up to 25.9, which were predominantly found in the GI tract. An exposure from mussels was calculated from an average portion of 225 g of mussel soft tissue containing four MP particles per gram. This would result in ingestion of 900 particles or 7 µg MPs per meal [9]. Considering the average consumption of marine fish, crustaceans, and mollusks per capita per year, combined with minimum and maximum numbers of particles per g for these three categories, annual ingestion per capita for marine fish, crustaceans, and mollusks has been estimated between, respectively, 25 and 32,375, between 322 and 19,511, and between 500 and 32,750 particles [12]. Notwithstanding these estimates, the major issues hampering the construction of a well-established view on exposure levels are the lack of a good definition of MPs, including details on size, shape, and composition, and the lack of harmonization of methods for detection and identification [13].

Additional data not included in the abovementioned reviews fit in the general view. Canned fish (sardine and sprat) obtained from a series of countries has been examined for the presence of MPs [15]. Fourteen out of 21 samples in this case did not show any MPs, with the other 7 samples showing counts ranging from 1 to 5 particles with a minimum size of 190 µm. The large size of the particles suggests improper gutting of the fish, since particles of these sizes are not expected to travel through the intestinal wall [15]. It is not clear whether the analytical method, based on extraction by filtration followed by micro-Raman spectroscopy, might have had an effect on the probability of detection of smaller MP particles. Mussels from eight coastal sites and eight supermarkets (fresh and processed) in the United Kingdom were investigated by Li et al. [35]. Total numbers between 0.7 and 2.9 MPs gram^−1^ tissue (wet weight (ww)) or between 1.1 and 6.4 MPs per individual mussel were found with a minimum size of 8 µm. Half of these particles were identified as MPs of the types PP and PET. Another 37% consisted of rayon and cotton, the latter partly a blend of cotton and olefin, and in all cases was interpreted as originating from anthropogenic sources [35]. Oysters of 17 locations along the Chinese coast were found to contained 0.62 (range: 0.14–2.35) MP particles gram^−1^ tissue (ww) or 2.93 MPs per individual oyster [36]. The minimum size as reported was 20 µm. The most common types found were cellophane, PE, and PET. Shellfish collected at 15 locations along the Pacific coast of Oregon (United States) showed concentrations of 0.35 (SD: 0.04) MP particles gram^−1^ tissue (ww) in oysters, and 0.16 (0.02) MP particles gram^−1^ tissue (ww) in razor clams were found [37]. The minimum size included was 63 µm. A majority of 95% were fibers, which were predominantly identified as PET and cellophane (CP). The MPs found were evenly distributed over gut and non-gut tissue [37]. These results fit with the assumption that higher counts can be expected with lower minimum sizes of the applied PSD due to the process of degradation of plastic particles. The overview of surveys of MP occurrence in shellfish carried out by Hantoro et al. does not include PSDs nor the lower limits applied [12].

Water, either tap or bottled, is of particular interest for human exposure. A study by the World Health Organization [22] has provided an overview of studies ranked according to a set of quality parameters. The application of a score for reliability of study design and results will be presented and discussed in the paragraph on quality assurance [20]. Tap water has shown lower concentrations of MPs compared to bottled water. The PSD is an important factor in comparisons of different studies. For example, studies by Oβmann et al. (bottled water) and Pivokonsky et al. (drinking water from surface water sources) both apply the same PSD (1 µm and higher) and a comparison revealed a 10-fold higher level of MPs in bottled water [38,39]. Other studies with a high score for reliability have applied different PSDs. The higher the lower limit of the PSD, the lower the number of particles that might be expected to be found [22]. In the following section some studies of bottled water with a high score for reliability will be reviewed.

A set of 259 bottles from 11 brands and 27 batches were analyzed by Mason et al. [34]. Identification of MPs was carried out using Nile Red staining and FTIR (for MP > 100 µm) or Nile Red staining with UV microscopy (for MP sizes between 6.5 and 100 µm) was performed and the results were corrected for background laboratory contamination levels using a series of blanks. Only seventeen bottles appeared to contain no MPs. An average of 10.4 MP particles larger than 100 µm per liter of bottled water was found. Particles with sizes between 6.5 and 100 µm were found with an average of 325 particles per liter. The maximum was found to be 10,390 MP particles per liter [34]. A set of 32 samples of bottled water from Bavaria (Germany) was investigated for the presence of MP particles of the size 1 µm and larger [38]. The counts of MP particles varied from 2649 ± 2857 (mean ± standard deviation) liter^−1^ in single use PET bottles and 4889 ± 5432 liter^−1^ in water from reusable PET bottles to 6292 ± 10521 liter^−1^ in glass bottles. Higher counts were found in older reused PET bottles than in new ones, but in all cases the diversity within each group was considerable. Particles smaller than 5 µm accounted for approximately 96% in the PET bottles and 78% in the glass bottles. In order of abundance, PET (76%) and PP (10%) were most commonly found in water from the PET bottles. In glass bottles, PE (46%), PP (23%), and a copolymer of styrene and butadiene (14%) were most common. Blank samples, based on ultrapure water, showed counts of 384 ± 468 MP particles liter^−1^. A comparable study was carried out by Schymanski et al. [40]. The number of particles per liter was considerably lower than that found in the study by Oβmann et al. [38]: 14 ± 14 (mean ± standard deviation) liter^−1^ in single use PET bottles, 118 ± 88 liter^−1^ in water from reusable bottles, and 50 ± 52 liter^−1^ in glass bottles. PE and PET were the most common types of MPs with total shares of 66% (glass bottles) and 83% (reusable PET bottles). Particles smaller than 5 µm were included in the study by Oβmann et al. but not in the study carried out by Schymanski et al. [38,40]. Calculation of the average number of particles larger than 5 µm, as reported in the Oβmann study, shows counts comparable to those found in the Schymanski study for PET bottles (Table 1). The counts for the glass bottles show a large difference, with the Oβmann study indicating much higher numbers of particles larger than 5 µm. The high share of the class of smaller particles fits with the assumed effect of particle degradation. Both authors mention as sources of MPs in glass bottles abrasion of bottle caps, bottling, and bottle washing machines and their washing liquids.

The results of the Oβmann study can be compared to the effect of water treatment in three water plants. The large share of particles being smaller than 5 µm was also found by Pivokonsky et al. [39]. Treatment was successful at removing 70–83% of the MPs, resulting in counts of 338 to 628 particles per liter on average in the resulting treated (drinking) water. The PSD of the MPs in these drinking waters was comparable to that found in the bottled waters. Concentrations of anthropogenic particles in 159 samples of globally collected tap water were reported by Kosuth et al. at an average level of 5.45 particles (range 0–61 particles) per liter [41]. The majority of these particles (98%) consisted of fibers with lengths between 0.1 and 5 mm. A large overview of studies on the presence of MPs in drinking water has been presented by Koelmans et al. [20]. Besides the four studies discussed above, most studies included in this review applied a lower size limit of approximately 20 µm. A large range of different types of plastic (with the highest shares being PE (28%) and PP (26%)) and of different particle shapes (with the highest shares being amorphous structures (35%) and fibers (25%)) were reported. Particles in the size range 3–500 µm were found in surface water in the Long Beach area (California, CA, USA) ranging from 2.8 to 18.5 particles liter^−1^ [42]. These numbers were established without further aid for detection. Following application of Nile Red staining for improved recovery, much higher amounts, from 45 to 701 particles liter^−1^, were found at the same locations. Although this study was not included in the Koelmans et al. review [20], several quality assurance measures were applied, such as appropriate sample size (20 liters), use of equipment with natural fibers only, contamination controls, positive controls for check of recovery, specificity of Nile Red staining, and control correction of the final results.

The release of NPs and MPs from tea bags made of nylon or PET fibers has been investigated by steeping four different brands in 10 mL water for 5 min at a temperature of 95 °C in triplicate [43]. The procedure used to detect particles was based on counting the number of particles and identifying their sizes using SEM and nanoparticle tracking analysis (NTA). For SEM, 100 µL samples were dried on silicon wafers and 90 images for the three triplicates per sample were collected and analyzed. The average number of particles calculated over the 90 images per sample was 1200 microparticles per mm^2^ and 7 million submicron particles per mm^2^. NTA was used to confirm the SEM count of submicron particles. Following extrapolation, the calculated amounts of particles per sample of 10 mL were approximated at 2.3 million microparticles and 14.7 billion submicron particles. Several quality assurance measures were taken, including steeping at room temperature, several controls, and the use of SEM for examining the state of the damaged fibers, showing the release of the microparticles. These numbers of particles are much higher than those reported for water in the already mentioned studies, indicating an increased presence of MPs, which is suggested to be due to the addition of the tea bags. However, this experimental approach did not match proposed sampling procedures, and the International Organization for Standardization (ISO) guideline for harmonizing sensory studies of tea was not followed [20,44]. This ISO guideline is currently in revision and re-publication is planned for December 2019. One major issue might be the cutting of the tea bags into small units, which might influence the release of MPs.

Comparison of human exposure to MPs via the food chain with other routes could add to a balanced evaluation. In their estimations for human exposure to MPs, Cox et al. included an approximation of exposure via air varying from close to zero to a maximum of 15 × 10^4^ microparticles per person per year [33]. SAPEA (Science Advice for Policy by European Academies) has reported an average air exposure of around 7.8 × 10^4^ microparticles per person per year [16], which is close to the estimate by Cox et al. Exposure via air can vary greatly depending on the type of sampling area (urban versus rural). A relationship between high versus low level exposure and the abundance of micro-sized conglomerates of airborne ambient black carbon particles has been found at the fetal side of human placentas [25]. Estimations of microparticles in air have been based on examination of snow from three regions in the Arctic and Europe [45]. The samples of snow were collected on ice in the Arctic sea (9 samples), Svalbard (5), Helgoland (2), Bremen, Germany (1), Bavaria, Germany (3) and the Swiss Alps (2). An average number of particles was found at 9.8 × 10^3^ per liter of melted snow. The highest concentration was found in a sample from Bavaria, Germany (15.4 × 10^4^ per liter). In this sample almost half of this number was contributed to by rubber. The lowest counts were found in the samples from the Arctic region with an average of 1.76 × 10^3^ particles per liter, with shares of rubber of up to 23%. The PSD ranged from 11 to 475 µm. The largest fractions were attributed to particles with a size of 11 µm (62%) and with a size of 11–25 µm (27%). The microparticle counts excluded microfibers, which were reported separately with 1.43 × 10^3^ fibers per liter on average in the European samples. The high frequency of rubber particles in the snow contrasts with the study by Klein et al. for Rhine sediment and the studies reviewed by Koelmans et al. for different types of water, where presumably rubber was excluded from the scopes of the methods [20,27].

Based on the 20 MP particles per 10 g of feces [6] and an average production of 128 g of feces per day per person [46], a discharge via feces of over 90,000 MP particles per year can be estimated. The remark by Cox et al. [33], that their estimated exposure to MPs from food (i.e., 39,000 to 52,000 particles per year depending on age and gender) is underestimated seems realistic.

## 3. In Vivo Uptake and Bioaccumulation Kinetics

An important issue is whether MPs can be absorbed and enter the systemic circulation. Various reports mention that particles 150 µm in size or smaller have the potential to cross the intestinal barrier of mammals, whereas particles 20 µm in size or smaller may have the potential to penetrate deeply into tissues [7,9,47,48]. Phagocytosis and endocytosis have been named as mechanisms for the uptake of particles smaller than 10 µm [16] and persorption in the Peyer’s patches of the ileum for particles up to 130 µm in size has been found, although a low uptake of 0.002% per 24 h was reported for the latter process [33]. The process of persorption has been extensively reviewed by Wright and Kelly [49]. EFSA reviewed the existing literature and concluded that intestinal absorption appeared to be very low, reaching up to 0.3% for particles of ~2–3 µm, based on rodent and human ex vivo models [9]. NPs are generally thought to have a higher potential to be absorbed than MPs due to their smaller size. An in vivo study in rats evaluating the biodistribution of 50 nm PS NPs after a single dose of 125 mg kg^−1^ bw through oral gavage has shown distribution of the NPs to various organs. The bioavailability was estimated in this case to be between 0.2 and 1.7% depending on the charge of the particles [50]. However, data on MPs are limited and both the EFSA and SAPEA reports stated that the uptake kinetics and distribution of MPs in human tissues are largely unknown [9,16].

### Rodent Studies

Since the publication of works by the EFSA and FAO, various new rodent studies have been published evaluating uptake and bioaccumulation kinetics and possible effects of MPs on tissues and metabolism [7,9]. An overview of these studies is given in Table 2. In all studies PS particles were used. In most studies the particles were indicated to be “pristine”, meaning that contamination with chemical compounds or pathogens from the environment could be excluded. However, contaminants resulting from their production and processing, such as monomers or softeners, might be present.

Two studies have reported on the uptake and bioaccumulation of MPs in mice. In a study by Deng et al., fluorescently labelled PS MPs of 5 and 20 µm were daily administered by oral gavage for 1, 2, 4, 7, 14, 21, and 28 days at a dose of 0.1 mg per day (corresponding to 1.46 × 10^6^ and 2.27 × 10^4^ particles of the 5 and 20 µm sizes, respectively) [51]. Wash-out groups were exposed for 28 days and sacrificed after a week without treatment. Using light and fluorescence microscopy, MPs were detected in various tissues. Liver, kidney, and gut tissue material was also digested using nitric acid and hydrogen peroxide, and fluorescence in these samples was measured to quantify MP concentrations. These measurements demonstrated gradual accumulation in all tissues that reached a steady state at around 14 days of exposure. After 28 days of exposure, concentrations in the liver, kidney, and gut for the 5 µm MPs were reported to be 0.077, 0.099, and 0.417 mg g^−1^ (ww), respectively, and for the 20 µm MPs 0.194, 0.082, and 0.234 mg g^−1^ (ww), respectively (levels based on wet weight were provided by Deng and Zhang [52]). After a wash-out period of 1 week, MPs could still be detected in all tissues (levels not reported). Comments on this study have been recently published by Braeuning [53] and Böhmert, Stock, and Braeuning [54] which were followed by a reply by Deng and Zhang [52]. One of the main issues was the high tissue concentrations measured. Based on the steady-state levels, Böhmert, Stock, and Braeuning [54] extracted an equation for an accumulation model and calculated an elimination half-life for the MPs of approximately 3 days. According to this model, the reported levels of MPs in the gut, liver, and kidney could only be reached when assuming a bioavailability close to 100%, and exclusive uptake by these organs, which are both unlikely. These results are clearly not in line with the EFSA report, in which intestinal absorption was suggested to be very low (≤0.3%), although this was based on limited studies [9]. Stock et al. have published the results of a 28 day trial with male HMOX-LacZ transgenic mice exposed to fluorescently labelled PS particles [55]. These mice contain a LacZ reporter gene, encoding for β-galactosidase, under control of the oxidative stress-responsive heme oxygenase-1. Mice were administered three times a week with a mixture of MPs of 1 and 4 µm (each 4.55 × 10^7^ particles/treatment) and 10 µm (1.49 × 10^6^ particles/treatment) by oral gavage. Animals were sacrificed three days after the last dosing [56]. Only a few MPs were detected in the intestinal cell layers but none were found in the liver, spleen, or kidney. The 72 h waiting time could have allowed at least some clearance between the treatments.

The large difference in tissue accumulation between the two studies cannot easily be explained by differences in exposure levels or by particle size differences. There are differences in the mice strains used, but also in the applied treatment scheme. In addition, it is unclear if the fluorescent label can be released from the PS particles in the animals or during the analysis, and how this could affect the results [57]. The large discrepancies in the results of both studies imply that more studies on uptake and bioaccumulation kinetics of MPs are needed to evaluate the risks of MPs in the food chain.

## 4. In Vivo Effects

When MPs enter tissues, potential effects might include physical stress and damage, apoptosis, necrosis, inflammation, oxidative stress, and immune responses [16,49,62,63]. Diffusion of monomers, toxic substances, or microbes from the particles to the surrounding tissue might contribute to chemical and microbiological hazards [7,16,49]. It is important to distinguish between physical and chemical induction of effects, since the mechanisms for these can be assumed to be principally different. An example of a physical effect is the so-called “frustrated phagocytosis”, which is the failure of macrophages to engulf their target and remove or destroy it, leading to a prolonged inflammatory process and possible tissue damage. This has been described in studies on the effects of carbon nanotubes (10–20 µm), asbestos fibrils (3–20 µm), and wear debris from implants [64,65]. Furthermore, in lung biopsies of workers in the synthetic textile industry, granulomatous lesions containing foreign bodies and interstitial fibrosis have been reported as an occupational hazard through inhalation exposure [49]. The type of effect has been indicated to be directly related to the shape of the particles [8].

Some papers describing the effects of MPs in animal models have been published, but the extent to which this can be translated to the human situation is under debate [7]. As already indicated above, translation towards oral exposure to MPs from food is especially challenging, since little is known about the dietary exposure and about the corresponding kinetics and biodistribution [16]. This section will explore the results of studies in which MPs were administered to mice which were published after the release of the reviews of EFSA and FAO [7,9].

### 4.1. Rodent Studies

Table 2 shows a total of six rodent studies that were published after the EFSA and FAO reports, focusing on the possible effects of MPs on tissues and metabolic pathways upon oral exposure [7,9]. Two of these studies, namely, the studies by Deng et al. and Stock et al., have also investigated bioaccumulation of MPs in tissues as discussed above [51,55]. The other four studies were all conducted by a research group of the Zhejiang University of Technology in Hangzhou, China.

In the study by Deng et al., unlabeled MPs of 5 and 20 µm were administered daily by oral gavage at concentrations of 0.01, 0.1, and 0.5 mg per day for 28 days [51]. Histological images showed inflammation and the presence of lipid droplets in the livers. It should be noted that little information was given on the histological evaluation procedure (e.g., the number of slides per tissue and endpoints etc.) and few images were shown, which makes it challenging to place the histological evaluation in perspective. Both the reported inflammation and lipid infiltration in the liver are not uncommon and can be seen in healthy animals. Hepatic levels of ATP, total cholesterol, and triglycerides decreased, as well as catalase activity, whereas the activity of lactate dehydrogenase, superoxide dismutase, and glutathione peroxidase increased. Metabolomic analysis of serum showed an effect of both particle sizes on levels of several metabolites and amino acids. These changes point in the direction of effects on energy and lipid metabolism, as well as oxidative stress.

The biomarkers were significantly altered versus the control animals, but no physiological ranges were discussed.

In the study by Stock et al., a mixture of MPs of 1 and 4 µm (each 4.55 × 10^7^ particles per treatment) and 10 µm (1.49 × 10^6^ particles per treatment) was administered three times a week by oral gavage for 28 days [55]. Animals were sacrificed 3 days after the last dosing [56]. Histological evaluation of liver, duodenum, ileum, jejunum, large intestine, testes, lung, heart, spleen, and kidneys revealed a tissue morphology without noticeable pathological findings. The use of HMOX-LacZ transgenic mice allowed evaluation of inflammation and/or oxidative stress, but no such effects were observed in different intestinal sections, liver, spleen, and kidneys. Again, for this study little information on the histological evaluation procedure was given.

The research group of the Zhejiang University of Technology (China) conducted a set of four studies with mice and examined various effects, including potential effects on the gut microbiome and on F1 and F2 offspring [58,59,60,61]. Exposure levels were comparable and the size ranges matched each other ([58,60]: 0.5 µm and 50 µm PS; [59,61]: 5 µm PS). The four studies differed among each other with regard to the set of evaluated endpoints, but the main reported effects included (1) an altered composition of the gut microbiota; (2) alterations in the intestinal barrier, demonstrated by decreased secretion of mucus and reduction in gene expression levels of ion transporters in the gut; and (3) modified lipid/fatty acid metabolism, as exemplified by changes in serum and liver triglyceride and total cholesterol levels, as well as gene expression and metabolomic profiling in the serum and liver (Table 2). Altered metabolism was also observed in the offspring. It should be noted that reduced body and liver weight were mentioned in the study by Lu et al. [58] but evaluation of these parameters was not mentioned in the subsequent studies. An important element in the study design was the unlimited availability of drinking water, which was used to administer the MPs. The water intake was not provided. This situation would have resulted in unknown and possibly variable amounts of consumed MPs. A reduced consumption of water due to, e.g., taste problems, could also underlie the effects on growth and liver weight.

The clear differences in the effects of oral MP exposure between the study by Deng et al. (and the Zhejiang University studies) and the study by Stock et al. seem in line with the observed differences in absorption of the applied MPs [51,55]. There are several possible explanations for these observed differences, including the use of mice strains with different genetic backgrounds (C57BL/6NTac versus ICR) and different treatment schemes (see above). Additionally, the use of possibly different procedures for the examination of histological changes and protocols for the analysis of effect parameters might have contributed to the discrepancy in the outcomes of the studies. These discrepancies require further evaluation or follow-up studies.

### 4.2. Supporting Documentation from Studies in Zebrafish

Exposure of zebrafish to MPs (PS; 5 μm; control, low (50 μg liter^−1^) and high (500 μg liter^−1^) exposure) has been found to result in distribution in the gut tissue, as evaluated by histological analysis [63]. Furthermore, inflammatory responses, oxidative stress, and changes in lipid metabolism in the gut tissue have been reported. The microbiota in the gut have also shown a modified composition. The low exposure dose has been found to result in a stronger response for many parameters compared to the high dose.

Raineri et al. used low density PE particles (sizes 125–250 µm) with or without an adsorbed amount of a mixture of polychlorinated biphenyls (PCBs), brominated flame retardants (BFRs), perfluoroalkyl substances (PFASs), and methylmercury [66]. Zebrafish were fed clean feed, feed with clean MPs, feed with MPs enriched with chemicals, and feed containing only the mixture of chemical compounds for three weeks. Levels of chemicals were highest in the liver, but the levels of MPs were not evaluated. The authors acknowledged that particles of the size range used are assumed to be incapable of migrating through the intestinal wall. Only livers of zebrafish exposed to MPs with the chemical mix showed white inclusions in the liver, which were not identified. Histological evaluation showed vacuolization in the liver for the group exposed to MPs with chemicals, but not for the group exposed to MPs only. The group exposed to chemicals only showed a low amount of vacuolization in the liver. Expression of genes coding for proteins involved in detoxification and oxidative stress was increased in the liver in both groups exposed to chemicals with or without MPs, but not in the MP only group.

## 5. In Vitro Experiments with Human Cells

Some in vitro studies with MPs using human cell lines have recently been documented. One of these is the study by Stock et al., who analyzed the uptake and effects of PS particles with sizes of 1 μm, 4 μm, and 10 μm in human in vitro systems [55]. To study the gastrointestinal uptake of the MPs, three different models with Caco-2 cells (human epithelial colorectal adenocarcinoma cells) were used, namely, a conventional monolayer of differentiated Caco-2 cells, a Caco-2/Raji-2 model containing M-cells (microfold cells) specialized in uptake, and a Caco-2/HT29-MTX co-culture model harboring mucus-producing goblet cells. The cells were cultured in a Transwell system containing a 3 μm pore size polycarbonate membrane. In all three models, particularly in the co-culture models, intracellular uptake was demonstrated for the 4 μm particles (maximum 3.8%) and to a lesser extent also for the 1 μm particles (up to 0.8%) and 10 μm particles (maximum 0.07%). The translocation of the 1 μm particles into the basolateral compartment of the Transwell system was examined and appeared to be less than 0.144%, which was the limit of quantification of the measurement method. Besides uptake, effects of increasing concentrations of the PS particles on the viability of proliferating Caco-2 cells were examined. Only exposure of the cells to the 1 μm particles at very high concentrations (>1 × 10^8^ particles per mL) resulted in a pronounced reduction of cell viability. To study the potential impact of the PS particles on the (intestinal) immune system, effects on PMA (phorbol-12-myristate-13-acetate)-induced differentiation/polarization of the human THP-1 monocyte cell line into macrophages were analyzed. The particles were found to not affect the differentiation and polarization of THP-1 cells. For confirmation of the in vitro findings, a 28 day oral feeding study was also performed (see Section 4 on in vivo effects).

Another in vitro study was performed by Hesler et al. [67]. They investigated the uptake, translocation, and effects of 50 nm and 0.5 μm carboxylated PS particles, as representatives for NPs and MPs, in Caco-2/HT29-MTX-E12 and BeWo b30/HPEC-A2 in vitro co-culture models for the human intestinal and placental barrier, respectively. Exposure of the in vitro models to both NP and MP materials did not lead to cytotoxicity or translocation but resulted in intracellular uptake of the particles. The particles were also tested for possible embryotoxic and genotoxic properties using the mouse ES-D3 differentiation (hanging drop) assay and micronucleus assay (CHO-K1 (Chinese hamster ovary) cells), respectively. Both particle sizes were found to show weak effects in the embryotoxicity assay, but were non-genotoxic.

Schirinzi et al. evaluated the effect of PE MPs (3–16 μm; accompanied by particles with sizes between 100 and 600 nm) and PS MPs (10 μm; accompanied by particles with sizes between 40 and 250 nm) on human T98G glioblastoma and HeLa cervical carcinoma cells. None of the MPs led to a significant reduction in cell viability [68].

Hwang et al. studied the in vitro effects of secondary (home-made) polypropylene MPs approximately 20 μm and 25–200 μm in size on various types of human cells, including peripheral blood mononuclear cells (PBMCs), dermal fibroblasts, the mast cell line HMC-1, the basophilic leukemia cell line RBL-2H3, the murine macrophage cell line RAW 264.7, and sheep red blood cells, using assays for cytotoxicity, reactive oxygen species (ROS) production, cytokine/histamine profiling, macrophage polarization, and hemolysis [69]. They found that the particles, irrespective of size, exhibited a relatively low effect on cell viability (using dermal fibroblasts and RAW 264.7 cells). However, a high concentration of the small sized particles increased the levels of cytokines (particularly IL-6 (interleukin-6) and TNFα) (tumor necrosis factor alpha) and histamines in PBMCs, RAW 264.7, and HMC-1 cells, suggesting a potential effect of these smaller particles on the immune system.

Finally, some studies are available which have engaged with micro (nano)plastics in the context of the use of plastics in prosthetics. These materials have also been reported to fragment, forming particles of micro- and nanosize, and to be able to translocate to the lymph nodes, and, in some cases, the liver and spleen. These studies have been recently reviewed by Oliveira et al. [62]. Two in vitro studies are mentioned in this review: exposure of RAW 264.7 cells to 2–3 μm ultra high molecular weight polyethylene (UHMWPE) and 1–10 μM polymethyl-methacrylate (PMMA) particles for 48 h led to the production of chemokines that were able to recruit THP-1 macrophage cells [70]. Treatment of PBMCs with 1 μm polyether-ether-ketone (PEEK) particles during 72 h resulted in low-level activation of Th1, Th2, and Th17 (T-helper) cells [71].

## 6. Methods for Detection and Identification

### 6.1. Elements in Methodology: Sample Preparation, MP Detection, and Identification

The procedures for the monitoring of MPs depend on the type of matrix. Evaluation of bioaccumulation in tissues of, e.g., animals in in vivo studies is principally different from monitoring products in the food chain, which might include feed. In this section monitoring of food and feed products, which can include the destruction of the sample matrix for the release of MPs, is evaluated. Basically, several steps are involved in the procedure from sampling of a certain amount and type of material up to and including the final result in terms of counts or concentrations, and are defined for the types of plastic included. The first step is the sample treatment and isolation of physical particles, which is followed by a detection technique and a final identification procedure [9,72]. In the following paragraphs these three methodological elements will be discussed. The next section will provide reflections on the quality assurance of methods.

#### 6.1.1. Isolation of MPs

A principle element is the sample preparation. MPs cannot be extracted from a sample by applying chemical solvents, whether polar (water) or apolar (organic solvents). The focus of sample preparation is primarily the destruction and/or removal of the matrix material rather than extraction of the target substance. Sample pre-treatment may range from no treatment (water) or grinding and sieving (compound feed and former food products) to enzyme digestion (soft animal tissues) or acid destruction (plant material and seaweed) [32,73,74]. For more severe treatments the effects on the different types of MPs need to be established. The ranges of polarity and of specific density of MPs are wide. Several types of MPs will dissolve in organic solvents and as a result will no longer be detectable. Additionally, the heat resistance differs among types, which limits the possibilities for sample destruction by heat treatment, either as a single parameter or combined with others.

MPs in organisms or food items can be released from the matrix using digestion of the sample matrix with acid, base, or enzymatic techniques, followed by isolation of the MPs from the digests [13,23,75]. The use of acid digestion is well known in analytical chemistry for trace element analysis and has also been used for the isolation of MPs [76]. Although a number of polymer types are resistant to such aggressive conditions, there are reports saying that the use of acid and to a lower extent base can result in damage or total destruction of plastic particulates. Such damage might include fusion of PS, PP, PET, low density PE (LDPE) and high density PE (HDPE), total loss of PA, and colour changes to other polymers [77]. Thiele et al. have published an overview of digestion methods assessed for their efficacy and damage to MPs [78]. Löder et al. have published an improved and universally applicable version of the basic enzymatic purification protocol that has been successfully applied to environmental samples with low concentrations of MPs [79].

Filtration is frequently used as a final step in the isolation of MPs from their matrix and depending on the digestion technique used, this may be more or less suitable. Filtration or sieving can be used to achieve size separation of MPs in water. Pore sizes of the most frequently used filters are between 0.2 and 500 μm. The smaller the pore size the more particles found, which not only reduces the size detection limit but probably confers the advantage that particles which can invade the functional system of mammals are sampled as well. In any case, the PSD, and most importantly its lower size limit, is a principle parameter of a method. In addition, difference in specific density between different types of MPs, or between a class of MPs and the matrix can be used for isolation of the particles. The specific density of MPs can vary considerably depending on the type of polymer and the manufacturing process. Density values for MPs range 0.8 to 2.3 g cm^−3^ [19,80], while typical densities for sand or sediments are in the range 2.60 to 2.94 g cm^−3^ [81]. This difference can be used to separate lighter MPs from sediment in a saturated NaCl or NaI solution (1.8 g cm^−3^) or in a solution based on sodium polytungstate (3.1 g/cm^3^). Plastics that float in fresh water and seawater are PS foam, HDPE, LDPE, and PP. The plastics that float in a high density sodium polytungstate solution include flexible and rigid polyvinyl chloride (PVC), PET, and nylon [13]. However, the buoyance force is only effective from particle sizes of approximately 1 µm and higher [26]. With respect to density separation, it should be noted that the specific density of several clay minerals is modified after being submerged in water. Most notably, the specific density of hydrated montmorillonite is reported to be as low as 1.40 g cm^−3^ [81], which could result in a lack of sedimentation for clay particles larger than 1 µm in solutions with a specific density higher than 1.40 g cm^−3^.

#### 6.1.2. Detection Techniques

Estimation of the numbers of particles of MPs is typically based on a microscopic technique. Accuracy and precision are dependent on the type and visibility of the particles. Several approaches have been developed to enhance the detection of the particles. A frequently applied technique is staining with Nile Red. This choice for a specific dye has been based on testing a range of alternatives, combined with several solvents [17,82,83]. Nile Red is solvatochromic, resulting in a red-shift of its fluorescence emission spectrum related to an increasing polarity of the material surrounding the Nile Red molecules. Different types of MPs show different responses when using a set of fluorescence filters (Figure 1). The limited emission window of the FITC/Rhodamine fluorochromes filter set (510–550 nm) passes less light with a higher dielectric constant of the targeted MPs. The filter set for FITC/Acridine Orange has no upper cut-off for the emission window (510 nm) and can therefore show a shift from green to yellow and to orange with a higher value of the dielectric constant. With a higher cut-off value adjusted for the TRITC/Ethidium Bromide fluorochromes (575 nm), only an orange–red emission is visible. The emission is faint at lower values of the dielectric constant. These trends are mentioned and applied in the literature as an aid in detection and primary identification [17,34,64,82,84]. A relationship has been suggested between the level of emission and the specific density of the type of MPs by Erni-Cassola et al. [85]. This is an interesting point of view for the distinction between MPs and, e.g., chitin flakes [85] or rubber tire wear [86], but a relationship between the relative brightness of the fluorescence and the type of plastic is difficult to establish. The specificity of Nile Red has been further documented by Wiggin and Holland [42]. The stained material can be subjected to FTIR afterwards [17].

#### 6.1.3. Identification of MPs

FTIR and Raman spectroscopy have long been used for the analysis of polymers, and it is therefore obvious to use these techniques to identify MPs [87]. FTIR enables the identification of particles down to 10 µm and may be used in transmission or reflection mode. While transmission produces the best quality spectra, reflection is the easiest technique as the infrared light does not pass the filter with the MPs situated on its surface. However, reflection mode measurements may result in spectral deviations due to multiplicative light scatter effects which complicate the identification of polymer components. For unequivocal particle characterization and identification, an appropriate infrared spectral database needs to be set up using a suitable multivariate statistics algorithm and incorporating potential machine errors as a result of, for example, multiplicative scatter effects. A range of common polymers like PP, PE, and PET can be identified by this technique. Imaging of areas without any preselection can be applied by using FTIR microscopes with focal plane array (FPA) detectors [73,88]. Automated analysis of images by software can produce information about the identity, number, and sizes of the individual particles [89]. In contrast to FTIR, Raman spectroscopy uses sub-micron wavelength lasers as its light source, and, as such, is capable of resolving particles down to 1 µm [23,90]. The Raman system laser is focused on the sample and the spectrum is simply acquired by collecting the scattered light. A drawback is that some samples exhibit fluorescence when irradiated by a laser which can obliterate the useful analytical Raman signal. A potential solution is to work at higher excitation wavelengths (for example, 1064 nm), although no reports have been found on the identification of MPs using this type of Raman laser. As with FTIR, Raman microscopy offers a choice of automation options, from simple discrete particle analysis to high-speed imaging [91]. Recently, the development of Raman Tweezers has been described as a tool for the identification of small and sub 1 µm MPs [92].

In addition to spectroscopic methods, another type of chemical identification is thermal analysis. Pyrolysis gas chromatography coupled with mass spectrometry (Py-GC-MS) has been used to identify MPs from different matrices based on their thermal degradation products. In contrast to reflectance spectroscopy, which results in profiles of the surface of particles, Py-GC-MS produces results of the whole particle. The advantage is the lack of interference from additional components such as pigments [93].

### 6.2. Quality Assurance and Quality Control

In view of the presence of MPs in the air and tap water, precautions in terms of quality assurance and control are elementary criteria for achieving reliable results. The need for harmonized protocols has been addressed by [12,13,20,94], among others. Parameters mentioned in these studies include blank samples for checking background levels, prevention of cross contamination among samples, reproducibility, harmonization of PSD, description of type and shape of the particles found, and conversion of counts to weight percent. These parameters belong to three different sets of quality assurance criteria: (a) method validation, (b) work flow, and (c) specific additional requirements for MP detection and identification. These three sets will be discussed in the following paragraphs.

Validation of analytical methods is a very common practice in the monitoring of food and feed. Decision (EC) 2002/657 provides principles for method validation [95]. Further guidelines have been published by several organizations covering the domain of analytical chemistry, such as the International Standard Organization (ISO), AOAC International, and the International Union of Pure and Applied Chemistry (IUPAC). The lists of parameters basically point to two basic requirements: accuracy (trueness, level of detection, and selectivity) and precision (measurement uncertainty, repeatability, and reproducibility). The derivation of the data for validation and the a priori criteria have been pointed out in several guidelines, such as the IUPAC Recommendations [96], the IUPAC Technical Report [97] and several AOAC guidelines (i.e., [98]). However, one principle step in MP monitoring is the counting of MP particles, which is in principle not part of analytical chemistry. As a consequence, the criteria as established for analytical chemistry cannot be adopted without evaluation of applicability or without modification. One issue is that measurement uncertainty has to be evaluated differently for measuring methods and for counting methods. A quality guideline for visual and microscopic monitoring methods is currently in development by Wageningen Food Safety Research in cooperation with the International Association for Feedingstuff Analysis (IAG) section Feed Microscopy, adopting relevant elements from other disciplines such as microbiology and molecular biology.

After validation of a harmonized method, the means of application and the work flow need to be designed. In general practice, negative controls (blank samples) and positive controls need to be examined at frequent intervals, for example biannually. Regular checks of air traps and participation in interlaboratory studies are usually part of laboratory quality systems. Basic requirements for accreditation of control laboratories have been laid down in an ISO guideline [99]. A first set of parameters for the control of work flow was developed by Thompson and Wood [100]. Unintentional contamination of test samples by the target from air, clothes, process water, and buffers, etc., is a major concern in molecular biology. Precautionary methods have been developed and need to be applied at the level of laboratory organization and at the level of runs of test samples [101,102]. A set of requirements for MP detection and identification has been developed by Hermsen et al. and Koelmans et al. [20,103]. This set has been used as a basis for testing the reliability of studies on water included in the WHO report [22]. Nine parameters are included in this set: sampling methods (avoidance of contamination, representative sample) with sufficient meta data on each sample, sample volume for achieving an appropriate level of detection but dependent on the particle size, sample processing and storage (conservation of samples), laboratory preparation (avoidance of contamination from coats and equipment), clean air conditions (clean room or laminar flow cabinet), negative controls, positive controls, sample treatment (digestion if necessary), and polymer identification [20]. Several of these requirements are already known and are applied in laboratory procedures. Background levels are known for, e.g., process contaminants (dioxins) which render the use of control samples necessary. Other requirements, such as clean air conditions and prevention of contamination by laboratory coats or plastic Eppendorf tubes are extra measures which are not principally installed for chemical methods.

There are specific elements in the procedures for MP monitoring. One element is a separate procedure for identification of polymers. The identification of compounds is a intrinsic part of analytical chemical methods. Signals resulting from mass spectroscopes are identified using libraries, and information on specific compounds is derived from these signals. Normal practice for visual methods is identification by the microscopic technician, supported by expert systems and written documentation. MP monitoring provides an opportunity to combine microscopic detection with a system for identification connected to a library with profiles. A major issue is the scope of the method in terms of the types of polymers to be monitored. A large array of polymers and particle shapes has been reported by the studies included in the review by Koelmans et al. [20]. The types of polymers to be included or the scope to be defined relate to the more basic requirements of analytical methods. AOAC recommends the application of an inclusivity and an exclusivity panel for analytical methods [98]. These panels are lists of target and non-target materials, to be considered and defined at the start of method development. Libraries for FTIR or Raman spectroscopy need to contain profiles for sufficient identification of targeted polymers as listed in the inclusivity panel. This is an important action for defining the scope of the method, since approximately 5000 types of plastic have been documented [28]. A selection of non-targeted polymers from the exclusivity panel needs to be included in the library as well, in order to prove that certain particles are correctly neglected in the final result for a sample. The same requirements apply to microscopic detection. Particles such as spheres, fibers, beads, film, flakes, and beads, etc., have been found in water [22], and the inclusion or exclusion of certain shapes in the scope of the method needs to be established. The same applies to the use of Nile Red as an aid for detection or identification. The incapability of Nile Red to stain, e.g., rubber does not automatically imply that rubber is to be listed on an exclusivity panel. The inclusivity panel, which also has to be defined for the step of microscopic detection, should include the appearance of the particles to be considered. The inclusivity and exclusivity panels as a priori definitions of the requested scope and the a posteriori performance of the method together provide the data for establishing the specificity or selectivity of the method as one of the quality parameters. Another principally important element of monitoring methods of MPs is the PSD. This type of documentation allows for comparison results of different studies, which is important for harmonization and for collecting results for large monitoring programs. A PSD also allows for the interpretation of the number of MP particles found.

As an overall view, monitoring of MPs requires a combination of expertise from different fields: methods can be assumed to consist of procedural steps from analytical chemistry and visual inspection. Elements for quality assurance from procedures in the domain of molecular biology require attention for their applicability in MP monitoring. This multidisciplinary approach is a challenge to design and to manage monitoring methods with sufficient levels of quality, reliability, and applicability.

## 7. Discussion

The SAPEA report has indicated that with equal or increasing exposure to MPs, a widespread ecological risk may arise in the next century [16]. A considerable volume of plastic is stored in the oceans, either floating or submerged, and this stock is waiting to be degraded to MPs and finally NPs [104,105,106]. There is currently no evidence of a widespread risk to human health from exposure to MPs. However, this situation should be identified as a data gap and not as the absence of a risk in view of the limited availability of data and only a selected set of types of MPs used in studies [16,22]. Recent research provides a diverse picture. Most of the exposure levels applied in experiments with human cell lines and mice are much higher than oral exposure levels of humans that can be deduced from the levels in some foods or water, which are considered to limit the relevance of the outcomes of these experiments [22]. Combination of exposure from different sources [6,12,33,34,38] would indicate a current human exposure exceeding 100,000 MP particles per capita, depending on the specific diet and geographic location, and on the particle size range included in the estimate. However, levels in food and water have been considered to be an underestimation. A comparison of exposure levels in in vivo studies, general exposure to humans from food and discharge via feces, and calculation of consumption from tap and bottled water as the source with the most reliable data, is given in Table 3. Two in vivo studies are excluded from these calculations because data are missing [60,61]. Furthermore, the data from Stock et al., who administered MPs three times a week, have been extrapolated to seven days per week, but it is unclear whether such a dosing scheme is representative of daily exposure [55]. Details and uncertainties in these studies are given in Table 2. Exposure in the in vivo studies shown in Table 3 in terms of the number of particles per kg body weight per day exceed the maximum estimations from bottled water. A comparison can be made between the exposure with the highest particle size (50 µm) used by Lu et al. [58] (146 # kg^−1^ bw day^−1^) and the estimated median of 256 particles kg^−1^ bw day^−1^ in stool (50–500 µm; [6]). Nevertheless, the total estimated exposure of humans from a general diet [33] and from analysis of human feces [6] should be considered to be underestimations, as has already been concluded by Cox et al. [33]. Explanations for this are the very incomplete data availability for food sources (15% of the total diet, [33]) and the high lower limit of the PSD in human feces (50 µm; [6]). Exclusively average concentrations in either tap or bottled water will account for, or exceed the estimations made from food and feces. The reported concentration of MPs in steeped tea and the calculated exposure based on a consumption of two cups per day might be assumed to be overestimated [43].

There are two basic difficulties with comparing the different sources of exposure as shown in Table 3. The first one is the difference between a fixed size of MPs as used in the in vivo studies compared to the range of sizes to be expected in surveys, even in the view of a Gaussian distribution of the sizes in pristine MPs in samples produced for a specific purpose. The second issue is the effect of using different lower limits of the PSD as applied in different surveys. As explained in the evaluation of some shellfish studies [35,36,37] and of surveys of bottled water (Table 1), a relationship can be expected between counts of MPs recovered and the lower limit of the PSD, due to the degradation of larger plastic particles to a higher number of smaller particles. The dominant occurrence of MPs at the lower end of the PSD is illustrated by actual data (Figure 2). The choice of the lower limit will very much influence the counts to be expected. The class of MP sizes between 501 and 1000 µm seems to be overrepresented in Figure 2. In reality, Teng et al. used a pore size of 1 µm for filtration, but their data were based on a lower limit of 20 µm [36]. This means that the lowest class of 1–500 µm is underrepresented, which is shown by an alternative trend line ignoring the smallest class (1–500 µm). This trend predicts an estimated amount of 712 MPs for this smallest class. The lower limit as finally applied by Teng et al. [36] might be related to the technical lower limit in practice of 20 µm for FTIR. A relationship between size and counts has also been established for particles of animal origin [107]. Applying these reflections to the data as shown in Table 3, caution should be taken when comparing different studies, and the lower limits of PSDs should always be considered. There is an urgent need for method harmonization, which should include the definition of the PSD.

Underestimation of MP levels in food and water could also be due to other limitations in the current detection methods. For instance, a category of “unidentified” particles is not listed in the major reviews [19,20], nor in the studies reviewed here, although it is reasonable to assume that not every chemical composition of MPs could be identified due to incomplete coverage in the libraries for FTIR or Raman identification. Overestimation might result from the inclusion of particles for which it is disputed as to whether they are covered by the definition of MPs in the strict sense. Rubber particles from, e.g., tire wear have been found in snow samples by up to 45% [45], but have not been reported in a set of over 40 studies on MPs present in water [20]. Despite this, emission of tire wear is reported to be a considerable source of MPs [11]. It has been argued that particles such as coating and tire wear should be included in the definition of MPs [10]. In studies focusing on particles of anthropogenic origin (e.g., [41]), rubber should have been included, but it was not mentioned. These procedural observations indicate that the design of inclusivity and exclusivity panels are of primary importance for all monitoring methods.

Another point to consider when monitoring MPs is the use of detection units. In all cases particle counts are only relevant if the particle size is given, since only the combination of size and number will eventually provide the opportunity to relate the observation to weight percentages. Expression in terms of parts per million (ppm) or parts per billion (ppb) has no significance for physical entities since these terms are defined for concentrations in the chemical domain. Particle size is also relevant for the discussion of whether MPs might be capable of crossing the intestinal border.

The approach for evaluating the effects of physical traumas and their mechanisms needs attention as well. Exposure to physical entities is hypothetically different from chemical compounds or nanoparticles due to different kinetic mechanisms [26]. The spatial dispersion of single invasions is to be assumed to follow a Poisson distribution [108,109], which might have a consequence for the evaluation of kinetics and dispersal of MPs. The option of locally induced agglomeration of nanoparticles to micro-sized particles, as observed for ambient black carbon particles in placentas [25] or for silicon aggregates from implants [110], complicates predictions and in silico modelling. This notion is recognized in the domain of visual research and microscopy [111]. The mechanism that a local physical trauma can result in a biochemical or physiological effect is not yet understood. It can be assumed that the primary response is at the level of cells or tissues, and histopathological examinations might be needed. Although exposure to MPs implies a combination of physical, chemical, and microbiological elements in a range of cases, the special situation of physical traumas needs specific attention.

Results obtained in mice studies by Deng et al. and Lu et al. [51,58], including related studies by the same research group, seem to be contradicted by those of Stock et al. [55], although differences in the schemes of administration and evaluation have been noted and might offer part of an explanation (Table 2). Complementary to this, the results from in vitro studies have indicated limited responses. Effects of long-term exposure, i.e., exposure which is longer than the four to five weeks which have now been applied, are unknown. Essentially, the in vivo studies point towards a set of diverse effects without a clear connection to the MP exposure levels, in terms of a quantified relationship. It has not yet been established as to whether a dose–response relationship will exist for MPs. In terms of a qualitative observation, certain effects in mice, most notably lower mucin secretion, modified gut microbiota, oxidative stress, and modified lipid metabolism, have been reported. These effects have been observed in zebrafish as well [63,66] and are supported by reviews [16,49,62]. These qualitative observations might be connected to oral exposure to MPs, unless alternative explanations exist. For example, a reduced body weight might be an experimental artefact due to reduced water consumption caused by the palatability of the MP particles [58]. Certain effects of MPs, in a qualitative sense, can be predicted based on general known biological processes. These mechanisms include disrupted phagocytosis, endocytosis, apoptosis, and persorption [8,49,62,63].

Parallel to an effect analysis, it has to be concluded that there is an urgent need for qualified monitoring methods for a range of food and feed materials in order to fill major gaps in the data on human exposure to MPs [13]. For water, which is an important part of MP exposure through food but which is usually excluded from the food production chain, a track record for documented methods and content levels is under construction [20]. Data on (long-term) absorption kinetics, biodistribution, and effects are necessary to achieve further insights into putative risks. Besides the need to have data on long-term exposure, the extrapolation from experiments based on animals from one type or strain and one developmental stage exposed to one type and shape of MP (currently only PS globules in reported in vivo studies with mice) to the diversity of real-life situations is difficult or is subjected to major uncertainty, and will provide a biased view on the topic [7].

These conclusions can be translated to a set of recommendations. Experiments for establishing dose–response relationships, if any, are needed. Harmonized protocols are needed for in vivo and in vitro studies. The existing OECD (Organization for Economic Co-operation and Development) guidelines for these types of studies need to be extended with specific elements related to MP exposure. These experimental parameters are not necessarily identical or comparable to those for nanoparticles, although parallels may be expected. Development, harmonization, and validation of monitoring methods covering a diverse array of feed and food material need to be achieved. A coordinated approach for all recommended activities, e.g., in an EU framework, is necessary.

## Figures and Tables

**Figure 1 foods-09-00072-f001:**
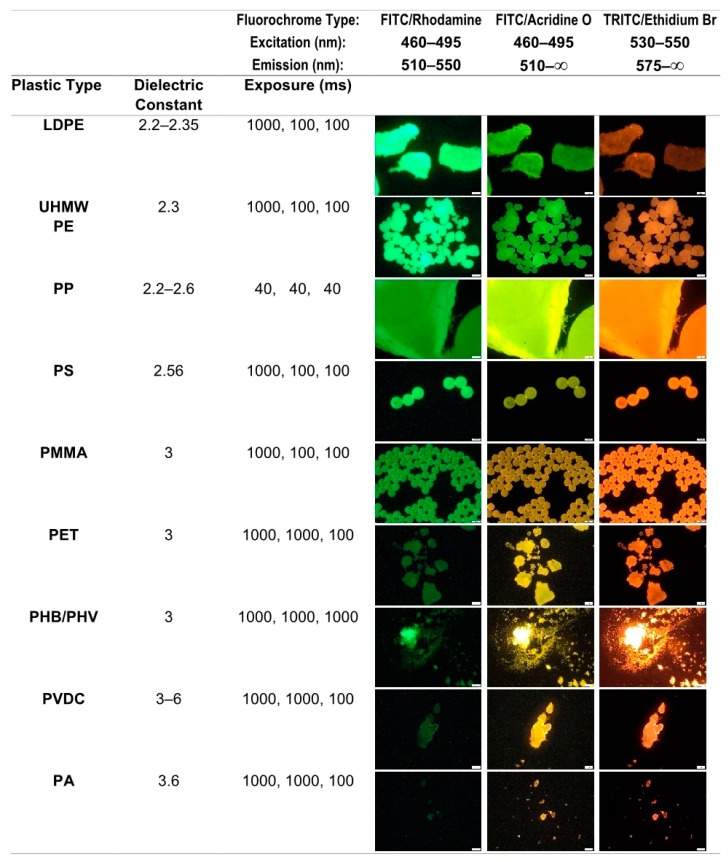
UV fluorescence images of nine types of plastic with three sets of excitation and emission wave lengths after staining with Nile Red in hexane solvent. The images were taken after full evaporation of the solvent. Exposure in terms of microseconds is indicated for proper comparison of the images. The types of plastic are ordered along their dielectric constant. Legend: PHB/PHV: polyhydroxybutyrate/polyhydroxyvalerate biopolymer; PVDC: polyvinylidene dichloride. Courtesy of Naomi Dam, Wageningen Food Safety Research.

**Figure 2 foods-09-00072-f002:**
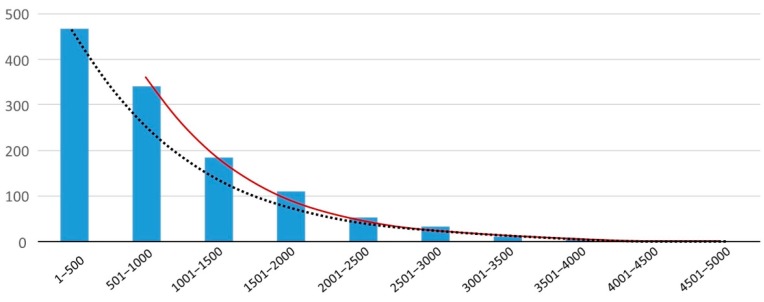
Distribution over size classes of the total set of examined MPs in oysters collected from the East Coast of China. The black dotted line represents a correlation of *r*^2^ = 0.972. Redrawn with permission from Elsevier from [36]. An alternative trend (red line) has been calculated ignoring the assumed underrepresented smallest class (*r*^2^ = 0.987).

**Table 1 foods-09-00072-t001:** Average numbers of particles per liter (*n*) and the standard deviation for water from three different types of bottles as reported in two studies. The study by Schymansky et al. only included particles larger than 5 µm. Legend: PET, polyester.

Bottle Type	Oβmann et al. [38]			Schymansky et al. [40]	
	*n* ± SD	% _> 5 µm_	*n* _>5 µm_	*n* ± SD	Min–Max Range
Single use PET	2649 ± 2857	1.7%	45	14 ± 14	2–44
Reusable PET	4889 ± 5432	4.6%	224	118 ± 88	28–241
Glass	6292 ± 10521	22.3%	1403	50 ± 52	4–156

**Table 2 foods-09-00072-t002:** Overview of in vivo rodent studies into the bioaccumulation and effects of microplastic. Study identification, design, methodology, results and comments are given. Legend: MP, microplastic; PS, polystyrene.

Study, Type ^a^ and Age of Animals Used	Experiment	Exposure Scheme	MP Type, Concentration	Tissues, Readout	Reported Result	Comments
Deng et al., 2017 [51]: ICR mice, aged 5 weeks	Bio-accumulation Effects	Oral gavage, 0.1 mg per day, 28 days and a wash-out group of 7 days Oral gavage, 0.01, 0.1, and 0.5 mg per day, 28 days	Pristine fluorescent PS MPs: 5 µm: 1.46 × 10^6^ particles per day; 20 µm: 2.27×10^4^ particles per day Pristine fluorescent PS MPs: 5 µm: 1 × 10^5^, 1 × 10^6^, and 5 × 10^6^ particles per day; 20 µm: 2 × 10^3^, 2 × 10^4^, and 1 × 10^5^ particles per day	Liver, kidney, and gut; fluorescence spectroscopy after 1, 2, 4, 7, 14, 21, and 28 days of exposure. Liver: histology. Serum: biomarkers and metabolomic analysis	Accumulation in all tested organs of both MPs. MPs were still present in all tissues after a wash-out period of 7 days. Liver inflammation and presence of lipid droplets. Disturbance of energy and lipid metabolism, oxidative stress, and neurotoxic responses	Unclear whether the gut was washed before measurement. Measured MP levels below standardized calibration curves. High accumulation; results point toward 100% bioavailability
Stock et al., 2019 [55]: Male HMOX1 reporter mice (C57BL/6NTac), aged 16–20 weeks	Bio-accumulation Effects	Oral gavage, mixture of three MP sizes, approximately 1.25, 25, and 34 mg kg^−1^ bw for 1, 4, and 10 µm particles, three times per week for 28 days. Animals sacrificed 3 days after last dosing. Similar to bioaccumulation	Carboxylated fluorescent PS MPs (1 µm), PS MPs (4, 10 µm). Per treatment: 1 µm: 4.55 × 10^7^ particles; 4 µm: 4.55 × 10^7^ particles; 10 µm: 1.49 × 10^6^ particles	Intestine, liver, spleen, and kidney: fluorescence microscopy. Duodenum, ileum, jejunum, large intestine, liver, testes, lung, heart, spleen, and kidney: histology (H&E staining and β-galactosidase staining)	Few MPs in intestinal cell layers, no MPs in liver, spleen, and kidney. No evidence of inflammation and/or oxidative stress (no induction of β-galactosidase expression)	Administration was 3× per week and animals were sacrificed 3 days after last dosing, so (some) clearance and recovery between and after last exposure would have been possible. Mice were older than in the other studies and had a different genetic background
Lu et al., 2018 [58]: ICR mice, aged 5 weeks	Effects	Exposure through drinking water (unlimited supply), 100 µg MPs per liter and 1000 µg MPs per liter, 35 days	Pristine MPs: 0.5 µm: 1.456 × 10^10^ particles per liter; 50 µm: 1.456 × 10^4^ particles per liter. Particle numbers correspond to the high dose	Colon: histology (mucus staining). Liver and serum: biomarkers. Microbiome composition (qPCR, sequencing)	Reduced body and liver weight for the high dose. Colon: reduced mucin excretion. Liver and serum: decreased serum indices (indicating decreased fat metabolism). Altered microbiota composition	Water intake not reported. Unknown amount of MP intake. Reduced body and organ weight at a high dose might be an experimental artefact
Jin et al., 2019 [59]: ICR mice, aged 5 weeks	Bio-accumulation Effects	Exposure through drinking water (unlimited supply), 1000 µg per liter, 42 days 100 µg per liter and 1000 µg per liter, oral gavage, continuous, 42 days	Fluorescent PS MPs: 5 µm: 1.456 × 10^7^ particles per liter. Pristine PS MPs: 5 µm: 1.456 × 10^6^ and 1.456 × 10^7^ particles per liter	Colon: fluorescence microscopy. Colon: histology (mucus staining), transporter protein expression. Liver, colon and, ileum: gene expression. Liver and serum: biomarkers. Serum: measurement of amino acids, carnitine, and succinylacetone. Bile acid: measurement of bile acids. Microbiome composition (qPCR, sequencing)	Presence of MPs in colon: decreased secretion of mucus, down-regulation of genes/proteins involved in ion transport. Altered amino acid and bile acid metabolism. Altered microbiota composition	Unknown amount of MP intake. Unknown amount of MP consumption
Luo et al., 2019a [60]: ICR mice, aged 7 weeks	Effects in offspring (F1)	Exposure through drinking water (unlimited supply) F0: 100 and 1000 µg per liter, exposure during gestation F1: no exposure	Pristine PS MPs: 0.5 µm: no particle concentration; 50 µm: no particle concentration. See Lu et al., 2018 [58]	F1 liver and serum: biomarkers. F1 liver: gene expression (fatty acid metabolism). F1 serum: measurement of amino acids, carnitine, and succinylacetone	Altered amino acid, carnitine, and fatty acid metabolism in the offspring without direct exposure to MPs	Unknown amount of MP consumption
Luo et al., 2019b [61]: ICR mice, aged 7 weeks	Effects in offspring (F1 and F2)	Exposure through drinking water (unlimited supply) F0: 100 µg per liter and 1000 µg per liter, exposure during gestation and lactation (approximately 42 days) F1: no exposure; offspring from F0 1000 µg per liter and control group used for production of F2 F2: no exposure	Pristine PS MPs: 5 µm: no counts. See Jin et al., 2019 [59]	Colon: histology (mucus and transporter staining). Liver: histology (H&E staining), biomarkers, and transcriptome analysis. Serum: biomarkers, measurement of amino acids, carnitine, and succinylacetone. Microbiome composition (qPCR, sequencing)	F0: altered gut barrier composition, altered hepatic gene expression, and modified gut microbiota. F1 (post-natal day 42): modified hepatic and serum metabolite levels, gut microbiota not altered. F1 (post-natal day 280): potential effects on lipid metabolism. F2 (post-natal day 42): few effects	Unknown amount of MP consumption. Assumed modification of the glycolipid metabolism is hypothetical. Only dams evaluated for some parameters

^a^: abbreviations used: ICR: Institute of Cancer Research; C57BL/6NTac: Taconics C75 black 6; HMOX1: LacZ reporter gene, encoding for β-galactosidase, under control of the oxidative stress-responsive heme oxygenase-1; bw: body weight; H&E: Haemotoxylin and Eosin staining.

**Table 3 foods-09-00072-t003:** Overview of exposure levels of MPs in mice studies, estimated exposure from diet and human stool, and drinking water, all expressed in particle counts. Particle sizes are provided for proper comparison. The data have been recalculated to give a daily exposure per kg body weight day^−1^. Data for bottled water in PET bottles have been used for calculations. The annual based data from Cox et al. and Schwabl et al. [6,33] were extrapolated to daily exposure. Studies are ordered along the increase of the lower limit of the particle size distribution (PSD) per section. For every particle size in the in vivo studies only the lowest exposure levels are shown. Average body weights: mice 40 g (based on [51]), humans 70 kg.

Material	Particle Size (µm)	Concentration (per L or Cup)	Estimated Daily Consumption	Exposure (Day^−1^)	Estimated Exposure (kg^−1^ bw Day^−1^)
Water, mice [58]	0.5	1.5 × 10^9^	4 mL	5.8 × 10^6^	1.5 × 10^8^
Gavage, mice [55]	1			4.6 × 10^7^	4.9 × 10^8^
Gavage, mice [55]	4			4.6 × 10^7^	4.9 × 10^8^
Water, mice [59]	5	1.5 × 10^6^	4 mL	5.8 × 10^3^	1.5 × 10^5^
Gavage, mice [51]	5			1.0 × 10^5^	2.5 × 10^6^
Gavage, mice [55]	10			1.5 × 10^6^	3.7 × 10^6^
Gavage, mice [51]	20			2.0 × 10^3^	5.0 × 10^4^
Water, mice [58]	50	1.5 × 10^3^	4 mL	6	146
Diet, food, maximum [33]	Depending on source				142
Diet, food and bottled water, maximum [33]	Depending on source				389
Stool, median [6]	50–500				256
Tap water, average [39]	1 and larger	470	2 L	940	13
Bottled water, average [38]	1 and larger	3.8 × 10^3^	2 L	7.5 × 10^3^	108
Bottled water, average [40]	5 and larger	66	2 L	132	2
Bottled water, average [34]	6.5–100	325	2 L	650	9
Bottled water, maximum [38]	1 and larger	1.7 × 10^4^	2 L	3.3 × 10^4^	475
Bottled water, maximum [34]	6.5–100	1.0 × 10^4^	2 L	2.1 × 10^4^	297
Tea per cup [43]	2.5 and larger	2.3 × 10^6^	2 cups	4.6 × 10^6^	6.5 × 10^4^

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
