# Peer review of "Current Insights into Monitoring, Bioaccumulation, and Potential Health Effects of Microplastics Present in the Food Chain"

_foods, 2020, doi:10.3390/foods9010072_

Round 1

Reviewer 1 Report

The Review with title "Current insights in monitoring, bioaccumulation, and potential health effects of microplastics present in the food chain" focuses on the concentration of microplastics (MPs) in food and water and their effect to human health. Particular attention was paid to the presence of MPs particles smaller than 1mm.

The Review is well presented and a big data set (more then 100 references) is available. Authors cited very recent published data, most of them from 2015 to 2019. However, this review will have a much bigger impact if a new paragraph about "reducing MPs" will be introduced. There are solutions (e.g. filtration, replacing plastic with natural products, biodegradable polymers, bioplastic, using green chemistry, washing) to reduce MPs in the food production and water ? If yes, are they applicable? There are actions that could be taken to prevent the bioaccumulation of MP in food chain or any other sources (e.g. air, sea, river)? Who and what will have the biggest impact in decreasing and prevent bioaccumulation of MP (e.g. industry, agriculture, human being)?  

Moreover some specific comments are listed below:

1. Keywords: The word "microplastic" is already present in the title, there is no need to repeat it again

2. Line 38-39: "Studies reported the presence of MPs in water, sea food including shellfish, sugar, honey, and beer." A Reference is needed here.

3. Line 108-108: "Primarily PE, polyamide (PA) and PP  was found." Replace "was" with "were"

4. Line 2018-219: "In glass bottles, PE (46%), PP (23%) and a copolymer of styrene and butadiene (14%) were most common. " Can you please explain why MP are present in glass bottles and where they come from?

5. Line 225: Please check the reference Oßmann et al., this author is spelled differently in the test an in the references.

6. Line 348: A full stop is missing between "scheme" and "The"

7. Line 466: A space is missing in the word "uptakof"

8. Line 496: there are 2 "in", delete one of them

9. Line 1006-1007: The reference number 71, it seems that will be published in 2020? Please double check the year of publication

Author Response

General remarks from the authors

We added a remark on the fluorescent label used in the animal studies (i.e. Deng et al), since we became aware of this issue quite recently. Therefore, to Line 354 was added: “In addition, it is unclear if the fluorescent label can be released from the PS particles in the animals or during the analysis, and how this could affect the results (Hoogenboom, personal communication with Zhang).” Some minor editorial changes were made throughout the text.

Reviewer 1 (RV1)

RV1: “The Review is well presented and a big data set (more than 100 references) is available. Authors cited very recent published data, most of them from 2015 to 2019. However, this review will have a much bigger impact if a new paragraph about "reducing MPs" will be introduced. There are solutions (e.g. filtration, replacing plastic with natural products, biodegradable polymers, bioplastic, using green chemistry, washing) to reduce MPs in the food production and water ? If yes, are they applicable? There are actions that could be taken to prevent the bioaccumulation of MP in food chain or any other sources (e.g. air, sea, river)? Who and what will have the biggest impact in decreasing and prevent bioaccumulation of MP (e.g. industry, agriculture, human being)?”

Authors response (AR)

AR: Thank you for your comment. The listing of methods for reducing MPs in food matrices is for now based on speculation and is therefore considered to be beyond the scope of this review, which considers risk analysis rather than MP management. Still we do acknowledge that a remark on this issue is useful. Therefore we suggest to add in Line 38 (introduction): “Methods for reduction of MPs are pending (i.e. filtration from wastewater effluent Talvitie et al. 2017), however insufficient data were reported to indicate trade-offs and effectiveness of those methods for food matrices in a meaningful way”

RV1: “1. Keywords: The word "microplastic" is already present in the title, there is no need to repeat it again

Line 38-39: "Studies reported the presence of MPs in water, sea food including shellfish, sugar, honey, and beer." A Reference is needed here. Line 108-108: "Primarily PE, polyamide (PA) and PP  was found." Replace "was" with "were" Line 218-219: "In glass bottles, PE (46%), PP (23%) and a copolymer of styrene and butadiene (14%) were most common. " Can you please explain why MP are present in glass bottles and where they come from? Line 225: Please check the reference Oßmann et al., this author is spelled differently in the test an in the references. Line 348: A full stop is missing between "scheme" and "The" Line 466: A space is missing in the word "uptakof" Line 496: there are 2 "in", delete one of them Line 1006-1007: The reference number 71, it seems that will be published in 2020? Please double check the year of publication”

AR: Thank you, we have adapted the issues raised accordingly in the respective manuscript lines.  

Considering remark 1: We would like to retain the word “microplastic” as a key word for the sake of manuscript exposure. Keywords, titles, summaries, etc. are categories in publication databases. There is almost always an overlap between titles and keywords. If someone searches for "microplastic" in a keyword database, he must be able to find our papers.

Considering remark 4: This remark regards Oßmann et al.  2018 and Schymanski et al. 2018. They present a number of different sources for the found MPs in glass bottles as the glass bottles themselves cannot be the source of MPs:

Abrasion of the bottle caps Bottling and bottle washing machines Washing liquids  

We have included a remark on this in line 234: “Both authors mention as sources of MPs in glass bottles abrasion of bottle caps, bottling and bottle washing machines and their washing liquids.”

Considering remark 9: This work of Corami et al. is published with the time stamp “Januari 2020” at the respective journal website of Chemosphere. The DOI however has a time-stamp of 2019: https://doi.org/10.1016/j.chemosphere.2019.124564 and is copyrighted by Elsevier in 2019. We suggest to keep the Journal reference: Fabiana Corami, Beatrice Rosso, Barbara Bravo, Andrea Gambaro, Carlo Barbante,

A novel method for purification, quantitative analysis and characterization of microplastic fibers using Micro-FTIR, Chemosphere, Volume 238, 2020, 124564.

Reviewer 2 Report

The authors present a review that deals with a new type of pollutant, namely microplastics (MPs). They touch several aspects of this subject, from the presence of MPs in the food chain, their possible health effects and the methodologies used to monitor them. The critical points and future developments are properly described.

I would suggest minor revision of the following point:

Many reviews talk about MPs in different contexts: I would suggest to report more clearly the points the differentiate and make this contribution original compared to the literature.

In Table 1 the standard deviations are very large and make it difficult to state in which plastic bottle there is the largest content of MPs: is this standard deviation around 100% typical for the quantification of MPs in water? In the references mentioned in Table 1, can it be figured out whether or not the material that forms the bottles influences the concentration/type of MPs?

In section 6.1.3, when drawbacks are mentioned (i.e. fluorescence for Raman), also some ways to overcome this problem should be identified (like working at longer excitation wavelengths).

The identification of analytes (in general) benefits from the possibility of carrying out the analysis in-situ with portable instruments: a quick mention that also portable Raman /FTIR instruments are nowadays available could be of interest for the reader.

If possible, the addition of an index would help to have a quick overview of the arguments tackled in this review.

Author Response

Reviewer 2 (RV2)

RV2: “Many reviews talk about MPs in different contexts: I would suggest to report more clearly the points the differentiate and make this contribution original compared to the literature.”

AR: Thank you for your comments. The review considers the latest reports (2015-2019) known on the presence, monitoring, bioaccumulation and potential health effects of MPs in the food chain after the 2016 and 2017 EFSA and FAO report. The review is therefore an original contribution as it is written from a food safety perspective. We have presented the scope of this review in the final paragraph of section 1 (Lines 79 – 89).

RV2: “In Table 1 the standard deviations are very large and make it difficult to state in which plastic bottle there is the largest content of MPs: is this standard deviation around 100% typical for the quantification of MPs in water? In the references mentioned in Table 1, can it be figured out whether or not the material that forms the bottles influences the concentration/type of MPs?”

AR: Oβmann et al. (2018) uses a particle cut-off value of 1 µm, whilst Schymanski used a cut-off value of 5 µm. Oβmann included pigmented particles and additives in his study, while Schymanski hardly addresses those issues. Both authors use micro-Raman spectroscopy to detect and identify MPs in bottled mineral water. It can be speculated that the high standard deviations found by both authors find its cause in the different brands and producers of the bottled waters tested. However, this was not further investigated.

It can be concluded that due to discrepancies between the findings of Oβmann and Schymanski in PET bottles (i.e. due to differences in water brands investigated), and the presence of large quantities of MPs in glass bottles (i.e. due to abrasion of the bottle caps, bottling and bottle washing machines, washing liquids), a direct link with the bottle material cannot be established.

RV2: “In section 6.1.3, when drawbacks are mentioned (i.e. fluorescence for Raman), also some ways to overcome this problem should be identified (like working at longer excitation wavelengths).”

AR: We have added to line 598: “A potential solution is to work at higher excitation wavelengths, for example 1064 nm, although no reports were found on the identification of MPs using this type of Raman lasers.”

RV2: “The identification of analytes (in general) benefits from the possibility of carrying out the analysis in-situ with portable instruments: a quick mention that also portable Raman /FTIR instruments are nowadays available could be of interest for the reader.”

AR: We acknowledge the presence of portable Raman and FTIR instruments on the market. However, those instruments are not equipped with a microscope extension and do not possess the accuracy and resolution to perform automated MP filter mapping (the current state of the art). The FTIR and Raman instruments are also cooled by means of liquid nitrogen, a feature which is not advisable to use on a portable instrument. As an addition for food and feed products, the sample preprocessing is not optimized for non-laboratory environments. Due to the high risk of MP contamination from non-sample related issues, it is not advisable to perform sample preprocessing on-site. 

RV2: “If possible, the addition of an index would help to have a quick overview of the arguments tackled in this review.”

AR: We will suggest this to the journal editor, as we are unaware if this is a common practice for Foods
